# A Closer Look at the Adversarial Robustness of Information Bottleneck Models

Iryna Korshunova [*]   David Stutz [* 1]   Alexander A. Alemi [2]   Olivia Wiles [3]   Sven Gowal [3]

## Abstract

We study the adversarial robustness of information bottleneck models for classification. Previous works showed that the robustness of models trained with information bottlenecks can improve upon adversarial training. Our evaluation under a diverse range of white-box $l_\infty$ attacks suggests that information bottlenecks alone are not a strong defense strategy, and that previous results were likely influenced by gradient obfuscation.

## 1. Introduction

The idea of an information bottleneck (IB) (Tishby et al., 1999) is to learn a compressed representation $Z$ of an input $X$ that is predictive of a target $Y$. This leads to the following training objective involving two mutual information terms:

$$\min_Z -I(Z, Y) + \beta I(Z, X). \quad (1)$$

This objective favours a representation $Z$ that retains the minimum amount of information about $X$ while being maximally predictive of $Y$. The hyper-parameter $\beta \geq 0$ controls the trade-off between the two losses.

To make this objective practical, Alemi et al. (2017) used variational techniques to construct the upper bound for the expression in Eq. 1 – also known as the Variational Information Bottleneck (VIB) loss:

$$\min_{p(\boldsymbol{z}|\boldsymbol{x})} \mathbb{E}_{p(\boldsymbol{x},\boldsymbol{y})p(\boldsymbol{z}|\boldsymbol{x})} \big[ -\log q(\boldsymbol{y}|\boldsymbol{z}) + \beta \log \frac{p(\boldsymbol{z}|\boldsymbol{x})}{q(\boldsymbol{z})} \big], \quad (2)$$

where $p(\boldsymbol{z}|\boldsymbol{x})$ is a stochastic encoder distribution, $q(\boldsymbol{y}|\boldsymbol{z})$ is a variational approximation to $p(\boldsymbol{y}|\boldsymbol{z})$, and $q(\boldsymbol{z})$ is the variational approximation to the marginal $p(\boldsymbol{z})$. In a similar way to the Variational Auto-Encoder (VAE) setup (Kingma & Welling, 2014), we can parameterize Gaussian densities $p(\boldsymbol{z}|\boldsymbol{x})$ and $q(\boldsymbol{y}|\boldsymbol{z})$ using neural networks, and fix $q(\boldsymbol{z})$ to be

---

[*]Work done during an internship at DeepMind [1]Max Planck Institute for Informatics [2]Google Research [3]DeepMind. Correspondence to: Iryna Korshunova <irene.korshunova@gmail.com>.

*Accepted by the ICML 2021 workshop on A Blessing in Disguise: The Prospects and Perils of Adversarial Machine Learning.* Copyright 2021 by the author(s).

a $K$-dimensional Gaussian $\mathcal{N}(\boldsymbol{0}, \boldsymbol{I})$, where $K$ is the size of the bottleneck layer. We can then use the reparameterization trick to learn the parameters of the neural networks when optimizing the stochastic estimate of the objective in Eq. 2.

A tighter bound on the IB objective is given by the Conditional Entropy Bottleneck (CEB) (Fischer & Alemi, 2020):

$$\min_{p(\boldsymbol{z}|\boldsymbol{x})} \mathbb{E}_{p(\boldsymbol{x},\boldsymbol{y})p(\boldsymbol{z}|\boldsymbol{x})} \big[ -\log q(\boldsymbol{y}|\boldsymbol{z}) + e^{-\rho} \log \frac{p(\boldsymbol{z}|\boldsymbol{x})}{q(\boldsymbol{z}|\boldsymbol{y})} \big], \quad (3)$$

where the second term uses a class-conditional variational marginal $q(\boldsymbol{z}|\boldsymbol{y})$, and $\rho$ is a hyper-parameter with the same role as $\beta$ in Eq.2. CEB parameterizes $q(\boldsymbol{z}|\boldsymbol{y})$ by a linear mapping that takes a one-hot label $\boldsymbol{y}$ as input and outputs a vector $\boldsymbol{\mu}_y$ representing the mean of the Gaussian $q(\boldsymbol{z}|\boldsymbol{y}) = \mathcal{N}(\boldsymbol{\mu}_y, \boldsymbol{I})$. CEB uses an identity matrix for the variance of $q(\boldsymbol{z}|\boldsymbol{y})$ and $p(\boldsymbol{z}|\boldsymbol{x})$, which is unlike VIB, where the variance of the encoder distribution is not fixed.

Multiple studies suggest that IBs can reduce overfitting and improve robustness to adversarial attacks (Alemi et al., 2017; Fischer & Alemi, 2020; Kirsch et al., 2021). For example, Fischer & Alemi (2020) showed that CEB models can outperform adversarially trained models under both $l_\infty$ and $l_2$ PGD attacks (Madry et al., 2018) while also incurring no drop in standard accuracy. However, no clear explanation has been found as to how IB models become more robust to adversarial examples. Previous works also failed to investigate possible effects of gradient obfuscation which could lead to a false sense of security (Athalye et al., 2018). In this paper, we continue the analysis into the behaviour of IB models in the context of adversarial robustness. Our experiments provide evidence of gradient obfuscation, which leads us to conclude that the adversarial robustness of IB models was previously overestimated.

## 2. Adversarial robustness

Since the discovery of adversarial examples for neural networks (Szegedy et al., 2014; Biggio et al., 2013), there has been a lot of interest in creating new attacks and defenses. In this section we briefly review methods for crafting norm-bounded adversarial examples. Later, we use these methods to assess the adversarial robustness of IB models.

The Fast Gradient Sign (FGS) attack (Goodfellow et al., 2015) is an $l_\infty$ bounded single-step attack that computes

an adversarial example $\boldsymbol{x}_{adv}$ as $\boldsymbol{x} + \epsilon \text{sign}(\nabla_{\boldsymbol{x}} \mathcal{L}(\boldsymbol{\theta}, y, \boldsymbol{x}))$, where $\boldsymbol{x}$ is the original image, $y$ is the true label, $\mathcal{L}$ is the cross-entropy loss, and $\epsilon$ is the perturbation size.

Projected Gradient Descent (PGD) (Madry et al., 2018) is the multi-step variant of FGS. The $l_\infty$ PGD attack finds an adversarial example by following iterative updates $\boldsymbol{x}^{t+1} = \text{Proj}_{\mathcal{B}(\boldsymbol{x}, \epsilon)}\left(\boldsymbol{x}^t + \alpha \text{sign}(\nabla_{\boldsymbol{x}} \mathcal{L}(\boldsymbol{\theta}, y, \boldsymbol{x}))\right)$ for some fixed number of steps $T$. Here, $\text{Proj}_{\mathcal{B}(\epsilon, \boldsymbol{x})}$ is a projection operator onto $\mathcal{B}(\epsilon, \boldsymbol{x})$ – the $l_\infty$ ball of radius $\epsilon$ around the original image $\boldsymbol{x}$. The attack starts from an initial $\boldsymbol{x}^0$ sampled randomly within $\mathcal{B}(\epsilon, \boldsymbol{x})$.

The reliability of PGD attacks often depends on the choice of parameters such as the step size $\alpha$, or the type of loss $\mathcal{L}$. Recent PGD variants are designed to be less sensitive to these choices, and it is common to run an ensemble of attacks with different parameters and properties. AutoAttack (Croce & Hein, 2020) and MultiTargeted (Gowal et al., 2019) are examples of this strategy.

## 3. Experiments

In this section, we experiment with VIB and CEB models on MNIST and CIFAR-10. We run a number of diagnostics, which indicate that gradient obfuscation is the main reason why IB models are seemingly robust. In trying to understand their failure modes, we also look at some toy problems. Our interpretation of the results is deferred to the next section. Hyperparameters of all our models and additional plots are included in the appendix.

### 3.1. MNIST

For VIB experiments on MNIST, we follow the setup of Alemi et al. (2017). Namely, for the encoder network, we use a 3-layer MLP with the last bottleneck layer of size $K = 256$. This bottleneck layer outputs the $K$ means and $K$ standard deviations (after a softplus transformation) of the Gaussian $p(\boldsymbol{z}|\boldsymbol{x})$. The decoder distribution $q(\boldsymbol{y}|\boldsymbol{z})$ over 10 classes is parameterized by a linear layer ending with a softmax. During training, we use the reparameterization trick (Kingma & Welling, 2014) with $S = 12$ samples from the encoder $\boldsymbol{z} \sim p(\boldsymbol{z}|\boldsymbol{x})$ when estimating the expectation over $p(\boldsymbol{z}|\boldsymbol{x})$ in Eq. 2. At test time, we also collect $S$ samples of $\boldsymbol{z}$, and compute $p(\boldsymbol{y}|\boldsymbol{x})$ as $\frac{1}{S} \sum_{s=1}^{S} q(\boldsymbol{y}|\boldsymbol{z}^s)$. We refer to this evaluation as *"stochastic mode"*. In the *"mean mode"*, we only use the mean of $p(\boldsymbol{z}|\boldsymbol{x})$ as an input to the decoder. Our deterministic baseline is an MLP of the same overall structure as the VIB model. We train it with a cross-entropy loss without any additional regularization.

First, we evaluate our models using the FGS attack. Figure 1 shows the robust accuracy of VIB models with varying $\beta$ under the FGS attack with different perturbation sizes $\epsilon$. For the rest of the paper, we assume that input images are

in the $[0, 1]$ range. Our results slightly differ from those of Alemi et al. (2017). In particular, the performance of our VIB models peaks at $\beta = 0.01$ instead of $\beta = 0.1$ as reported previously, and the evaluation in the "mean mode" and "stochastic mode" does not lead to the same results.

Despite these differences, we can still achieve large gains in robust accuracy under the FGS attack for VIB models in comparison to the baseline. One result that stands out is the unusually high robust accuracy under the attack with $\epsilon = 0.5$. Indeed, with this perturbation size, one can design an attack that makes all images solid gray and, as such, the classifier should not do better than random guessing (Carlini et al., 2019). The obtained robust accuracy above 10% indicates that gradients of VIB models do not always direct us towards stronger adversarial examples. To check if the improvements in robust accuracy generalize to stronger attacks, we evaluate VIB models with $\beta = 0.01$ under the PGD attack with 40 steps, $\alpha = 0.01$, $\epsilon = 0.2$, and a different number of restarts. Figure 2 shows that we can drive the robust accuracy to zero as we increase the number of restarts. It is an indication of gradient obfuscation, as the loss landscape cannot be efficiently explored by gradient-based methods (Carlini et al., 2019; Croce & Hein, 2020).

### 3.2. CIFAR-10

For CIFAR-10, as our encoder network we use a PreActivation-ResNet18 (He et al., 2016) followed by an MLP with the same architecture as the MNIST experiments. We train this network end-to-end, and only use random crops and flips to augment the data. As previously, we construct an analogous deterministic model that we do not regularize in any way, thus it overfits.

In Figure 3a, we evaluate the adversarial robustness of CEB models on $l_\infty$ PGD attack with 20 steps and $\alpha = 0.007$ (Madry et al., 2018). It is surprising that some of our deterministic models can outperform an adversarially-trained ResNet from Madry et al. (2018) with a reported robust accuracy of 45.8%. This result alone suggests that PGD attacks should be used with caution when evaluating models that might obfuscate the gradients. As with MNIST, we can again significantly reduce the robust accuracy by increasing the number of restarts as shown in Figure 3b.

To get a better estimate of the robust accuracy in the presence of gradient obfuscation, we use a set of stronger attacks: a mixture of AutoAttack (AA) and MultiTargeted (MT) (Croce & Hein, 2020; Gowal et al., 2019). We execute the following sequence of attacks: AutoPGD on the cross-entropy loss with 5 restarts and 100 steps, AutoPGD on the difference of logits ratio loss with 5 restarts and 100 steps, MultiTargeted on the margin loss with 10 restarts and 200 steps. From Figure 3c, we see that deterministic models

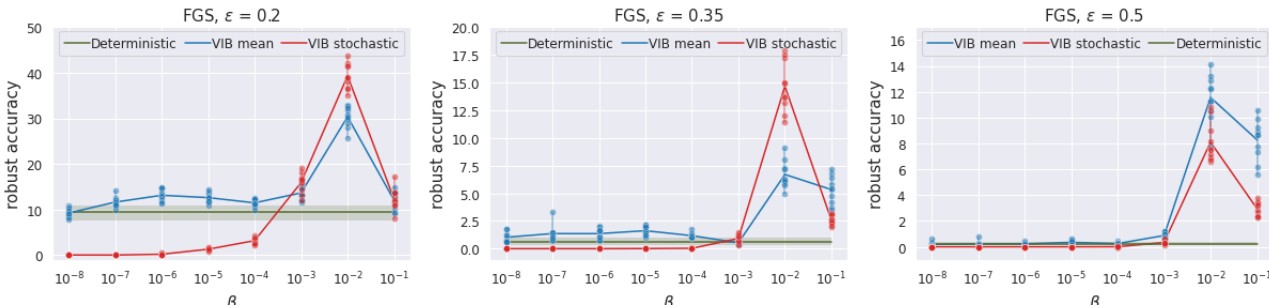

Figure 1. Robust accuracy under FGS attack with $\epsilon = 0.2, 0.35, 0.5$ on MNIST. VIB models are evaluated in two modes: stochastic and mean. For each $\beta$, we trained 10 models with different random seeds. Dots indicate the results of evaluating each individual model. For deterministic models, solid line plots the average robust accuracy over 10 models, while the hue gives the minimum-maximum range. The standard accuracy of all models is between 98.2% and 98.9%.

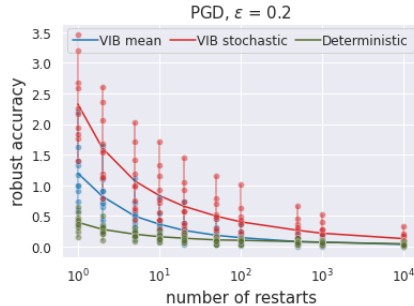

Figure 2. Robust accuracy of MNIST VIB models with $\beta = 0.01$ under a PGD attack with 40 steps and a step size of 0.01 for different number of restarts. At each restart, the initial point $x^0$ is sampled randomly within $\mathcal{B}(\epsilon, x)$.

have zero robust accuracy, while the performance of CEB models varies across models with different random seeds. This dependence on the seed could be the consequence of sub-optimal network initialization and difficulties related to training IB models. Some part of the variance in the robust accuracy might still be attributed to having an imperfect attack due to the unreliable gradients.

Finally, in Figure 4 and in the appendix, we show typical loss landscapes produced by the CEB model with $\rho = 2$ that scored 15.8% accuracy under the AA+MT ensemble of attacks. These plots are strikingly different from typical smooth non-flat loss landscapes obtained from adversarially trained models (Qin et al., 2019). The flatness of the plotted landscapes explains why gradient-based attacks with cross-entropy loss are not as effective. Moreover, since IB losses do not explicitly penalize misclassification for perturbed inputs within a certain $l_p$-ball, the model is free to choose where to place decision boundaries. Figure 4 suggests that CEB models could be robust to much smaller perturbation radii.

### 3.3. A toy problem

We established that gradient obfuscation makes it harder to understand the robustness properties of IB models on real

datasets. Thus, analysing toy examples can be a useful alternative. A classification task from Tsipras et al. (2019) is one example that can motivate the use of IBs, where their ability to ignore irrelevant features becomes helpful. We study this problem in the appendix. Here, we consider another simple setup where labels $y$ are sampled uniformly at random from $\{-1, 1\}$, and two features have the following conditional distributions:

$$p(x_1|y=1) = \mathcal{U}(0, 10), \; p(x_2|y=1) = \begin{cases} \mathcal{U}(0, 1) \text{ w.p. } 0.9 \\ \mathcal{U}(-1, 0) \text{ w.p. } 0.1 \end{cases}$$

$$p(x_1|y=-1) = \mathcal{U}(-10, 0), \; p(x_2|y=-1) = \begin{cases} \mathcal{U}(-1, 0) \text{ w.p. } 0.9 \\ \mathcal{U}(0, 1) \text{ w.p. } 0.1 \end{cases}$$

In this example, the label can be predicted from the sign of $x_1$, so in the optimal IB case, we need to communicate 1 bit of information about the input. The first feature is also more robust since it requires a larger perturbation before its sign gets flipped. In practice, we found that a simple VIB classifier does not exclusively focus on $x_1$, and so it becomes prone to a rather trivial attack that substracts or adds $\epsilon = 1$ to $x_2$ depending on the label, as shown in Figure 5. This could be the consequence of SGD training, the approximate nature of the objective function, VIB's formulation as a combination of competing objectives or other reasons we do not yet understand.

## 4. Discussion

By re-evaluating adversarial robustness of VIB and CEB models, we have shown that weak adversarial attacks are often unable to provide reliable robustness estimates as these models create highly non-smooth loss surfaces, which are harder to explore with gradients. Therefore, we believe that previous, as well as future results on the robustness of IB models should include basic checks for gradient obfuscation. This is especially important when comparing different types of models, e.g. IBs versus adversarial training.

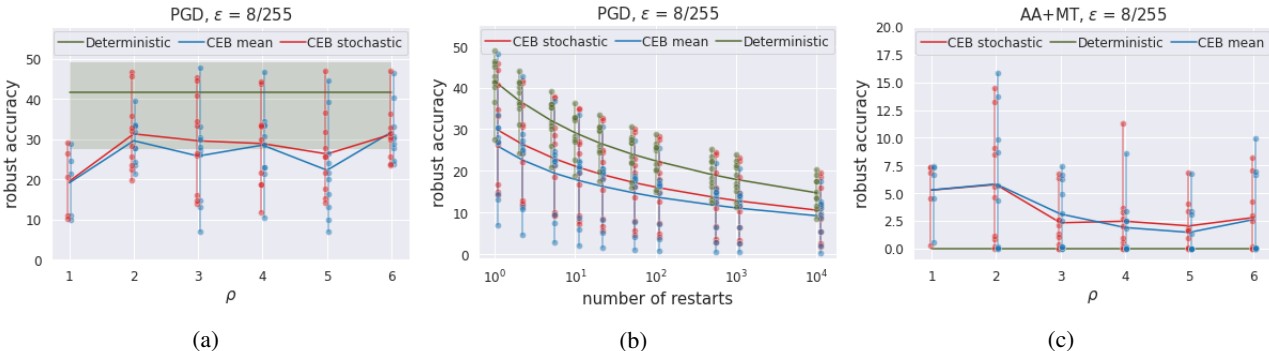

(a)  (b)  (c)

*Figure 3.* Robust accuracy of CEB and deterministic models on CIFAR-10 **(a)** Under the PGD attack with parameters from Madry et al. (2018) and a single restart **(b)** Under the PGD attack with different number of restarts, where we evaluated CEB models with $\rho = 3$. **(c)** Under the ensemble of AutoAttack and MultiTargeted. Each model was trained with 10 random seeds, and we excluded those runs where performance collapsed to a random chance, which happened mainly for the most regularized models with $\rho = 1$. Standard accuracy of the remaining models was above 90%.

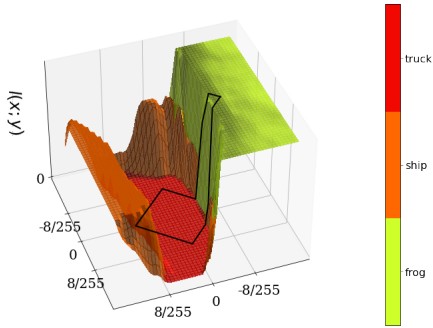

*Figure 4.* Cross-entropy loss surface produced by the highest-scoring CEB model (15.8% under AA+MT) for a test image of a truck. The diamond-shape represents the projected $l_\infty$ ball of size $\epsilon = 8/255$ around the original image. The surface is generated by varying the input to the model, starting from the original input image toward either the worst attack found using PGD or the one found using a random direction.

Our experiments were inconclusive as to whether IB models offer adversarial robustness gains relative to the undefended deterministic baseline. For MNIST, the results under the FGS attack seemed promising. However, looking at the performance under the PGD attack with multiple restarts and different perturbation sizes showed a different picture. For CIFAR-10, some of the CEB models were significantly better than the baseline under the strongest attack. However, we did not identify the exact cause for having excessive variance in the results of models with different random seeds. Thus, it would be interesting to find regimes where CEB can reliably converge to more robust models.

In this paper, we only considered IB models in discriminative settings. A generative model related to VIB is $\beta$-VAE (Higgins et al., 2017). For auto-encoders, the adversarial attack amounts to finding inputs that would cause the decoder to reconstruct a visually distinct image, e.g. an object from a different class. Camuto et al. (2021) showed that

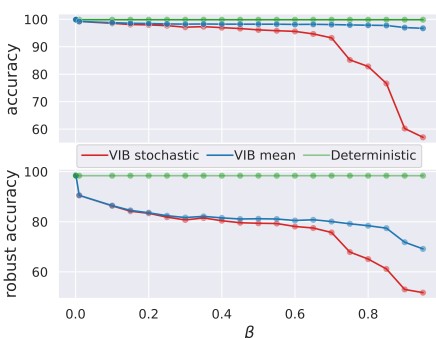

*Figure 5.* Standard and robust accuracy of a VIB and a linear deterministic classifiers on the toy problem from Section 3.3.

$\beta$-VAE for larger values of $\beta$ is more robust to adversarial attacks. However, Kuzina et al. (2021) used a different set of evaluation metrics to challenge this claim. Cemgil et al. (2020) attributes the lack of robustness of $\beta$-VAE models to the inability of their objective to control the behaviour of the encoder outside of the support of the empirical data distribution. Namely, without additionally forcing the encoder to be smooth, tuning $\beta$ alone is not enough for learning robust representations. Together with our observations for VIB and CEB models, the disagreement about $\beta$-VAE's results corroborates the need for more nuanced evaluation before adversarial robustness claims can be made.

Overall, we believe that using IBs in the context of adversarial robustness is an idea that deserves further exploration. In this paper, we focused on the empirical evaluation of IB models under standard robustness metrics and illustrating the caveats related to it. An interesting future research direction would be to understand the properties of IB models, especially in the stochastic regime, from both information-theoretic and adversarial robustness perspectives. Another promising direction would be to explore IBs with additional curvature regularization (Moosavi-Dezfooli et al., 2019; Qin et al., 2019) or in combination with adversarial training.

## Acknowledgements

We would like to thank Taylan Cemgil, Lucas Theis, Hubert Soyer, Jonas Degrave, and the wonderful people from the robustness teams at DeepMind for their help with this project, interesting questions, valuable discussions, and feedback on the paper.

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

# Appendix

## Toy example

In Figure 6, we plot 1K samples from the data distribution as outlined in Section 3.3. We train both a deterministic and a VIB model. For the deterministic model, we used a linear classifier whose weights and biases were initialized with zeros. For the VIB model, we used the bottleneck of size 2. The weights of the encoder were initialized with Xavier uniform scheme (Glorot & Bengio, 2010). The linear decoder's weights were initialized to zero. We use 12 samples from $q(\boldsymbol{z}|\boldsymbol{x})$ during training as well as for the stochastic evaluation mode. We optimize the parameters of both the linear deterministic and the VIB model using SGD with a learning rate of 0.003, momentum of 0.9 and Nesterov updates. We perform 1000 iterations with a batch size of 1024 (re-sampled from the data distribution each iteration) and the same random seed for both models. We evaluated clean and robust accuracy on a fixed set of 10K samples.

To this end, considering the discussion in (Stutz et al., 2019; Tsipras et al., 2019), we create a pre-computed set of adversarial examples by sampling exclusively from the low-density regions, cf. Figure 6. This emulates adversarial examples directly attacking the feature with weak correlation and is reasonable due to the low dimensionality, i.e., $d = 2$. It also means that we do not consider classical $l_\infty$ constrained adversarial examples. This is because, even for small $\epsilon$, such adversarial examples are not guaranteed to preserve the original label. This is also complementary to the toy example by Tsipras et al. (2019) discussed below.

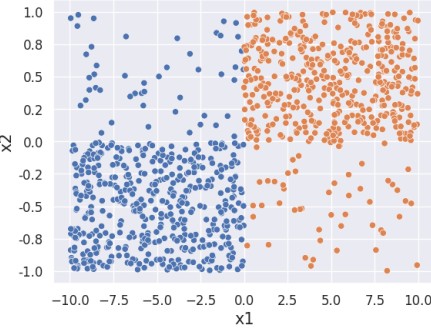

*Figure 6.* 1K random samples from the data distribution of our toy example discussed in Section 3.3. Colors indicate class label.

## Toy example from Tsipras et al. (2019)

For our second toy problem, we consider a binary classification task from Tsipras et al. (2019), where $y \sim \{-1, 1\}$ uniformly at random, and features are distributed as:

$$x_1 = \begin{cases} +y \text{ w.p. } p \\ -y \text{ w.p. } 1-p \end{cases} \quad x_2 \ldots x_{d+1} \overset{i.i.d}{\sim} \mathcal{N}(\eta y, 1). \quad (4)$$

We choose $p = 0.95$, $d = 100$, and $\eta = 0.3$. An adversarial attack with $\epsilon = 2\eta$ can shift Gaussian features towards the opposite class, so that $x_2 \ldots x_{d+1} \overset{i.i.d}{\sim} \mathcal{N}(-\eta y, 1)$. Thus, it becomes easy to fool a classifier that relies on these features. Note, however, that this might also change the true label according to the data distribution. Nevertheless, one might expect that IB models are more robust in this case since the compression cost forces to focus on $x_1$ – the feature that highly predictive of the label. Indeed, if we look at Figure 7, this seems to be the case, but oddly, only for the case of stochastic evaluation. Note that Tsipras et al. (2019) constructed this problem to demonstrate that clean and robust accuracy were at odds with each other, and this is what we also see in Figure 7. There are, however, doubts whether this toy example can reflect what happens in real-world scenarios (Yang et al., 2020; Stutz et al., 2019).

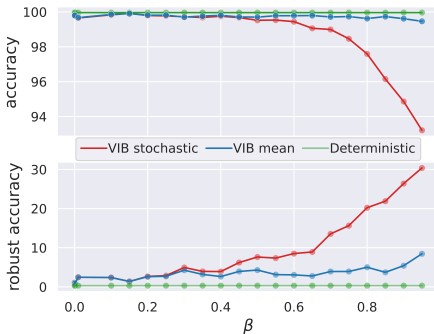

*Figure 7.* Clean and robust accuracy of a VIB and a linear deterministic classifiers on a toy problem from Tsipras et al. (2019).

On this toy example, we used the same setup as described above for our own toy example. However, we adapted the VIB bottleneck to be of size 25 due to the increased dimensionality, i.e., $d = 100$ and only perform 200 update steps.

## Architectures and hyperparameters

For MNIST experiments, we based our JAX (Bradbury et al., 2018) implementation on the original VIB code: `github.com/alexalemi/vib_demo`. We used the following MLP architecture for the encoder: 1024 - ReLU - 1024 - ReLU - $2K$, with $K = 256$. The decoder consisted of a single dense layer with a softmax nonlinearity over 10 outputs. All weights were initialized using the default Xavier uniform scheme (Glorot & Bengio, 2010), and all biases were initialized to zero. We used Adam optimizer with an initial learning rate of $10^{-4}$ and parameters $\beta_1 = 0.5$, $\beta_2 = 0.999$. We decayed the learning rate by a factor of 0.97 every 2 epochs. The batch size was set to 100, and we trained the networks for 200 epochs. Input images within a $[0, 1]$ range were rescaled inside the network to $[-1, 1]$ range prior to passing them to the first dense layer. We used Polyak averaging (Polyak & Juditsky, 1992) with a constant decay of 0.999. For 256 outputs from the bottleneck layer that correspond to the standard deviation of $p(\boldsymbol{z}|\boldsymbol{x})$, we used the following softplus transformation to make them positive: $\sigma(x) = \log(1 + \exp(x - 5.0))$. Our deterministic baseline models had the same overall structure as the VIB models, i.e. 1024 - ReLU - 1024 - ReLU - $K$ - 10 - softmax, with all training hyperparameters as above.

For CIFAR-10 experiments, the encoder network was a concatenation of a PreActivation-ResNet18 (He et al., 2016) with the same MLP as in our MNIST setup. The decoder $q(\boldsymbol{y}|\boldsymbol{z})$ and the backward encoder $q(\boldsymbol{z}|\boldsymbol{y})$ in CEB were again one-layer networks. We trained everything end-to-end for 1000 epochs. The batch size was set to 1024, and we used Adam with an initial learning rate of 0.012 and default $\beta$ parameters. The learning rate was multiplied by 0.3 every 250 epochs. For CEB, we annealed $\rho$ from an initial value of 100 down to its target value during the first 4 epochs. Similarly, for VIB, we increased $\beta$ from $10^{-8}$ to its target during the first 100 epochs. Prior to the first ResNet layer, input images within a $[0, 1]$ range were normalized using per-channel means and standard deviations computed across the train set of CIFAR-10.

## Additional results on MNIST and CIFAR-10

Below, we provide additional figures for the experiments in Sections 3.1 and 3.2. For MNIST, Figure 8 shows the results of increasing the number of restarts when we use a PGD attack with $\epsilon = 0.1$. For CIFAR-10, Figure 9 plots the robust accuracy of VIB models under various attacks, and Figure 10 illustrates cross-entropy loss surface of a CEB model on a couple of test images.

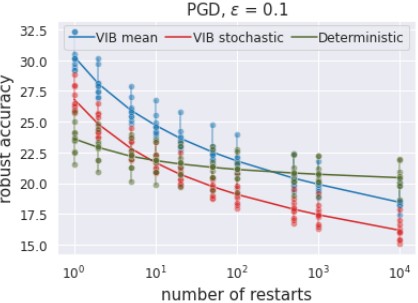

*Figure 8.* Robust accuracy on MNIST dataset of VIB models with $\beta = 0.01$ under a PGD attack with 40 steps, a step size of 0.01, and perturbation $\epsilon = 0.1$ for different number of restarts. At each restart, the initial point $\boldsymbol{x}^0$ is sampled randomly within $\mathcal{B}(\epsilon, \boldsymbol{x})$. Here, we see that VIB models can be made less robust than the deterministic baseline.

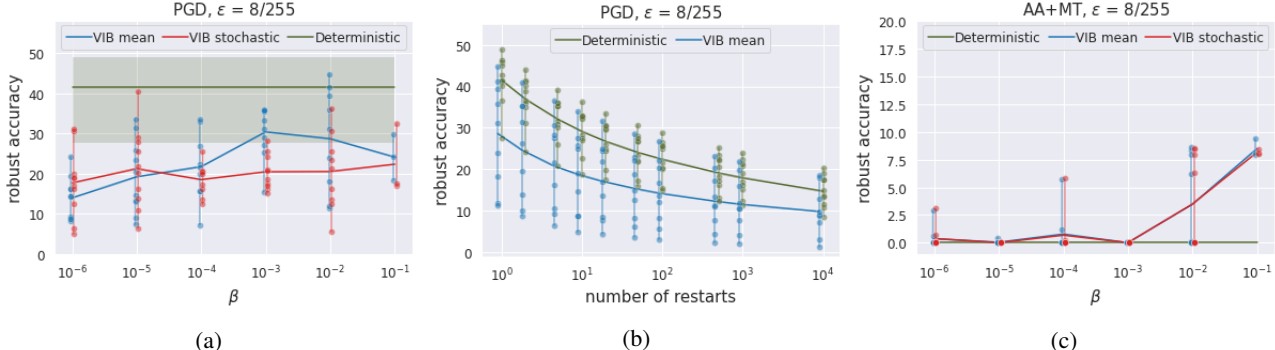

(a)  (b)  (c)

*Figure 9.* Robust accuracy of VIB and deterministic models on CIFAR-10 **(a)** Under the PGD attack with parameters from Madry et al. (2018) and a single restart **(b)** Under the PGD attack with different number of restarts, where we evaluated VIB models with $\beta = 0.01$. **(c)** Under the ensemble of AutoAttack and MultiTargeted. Each model was trained with 10 random seeds, and we excluded those runs where performance collapsed to a random chance, which happened mainly for the most regularized models with $\beta = 0.1$.

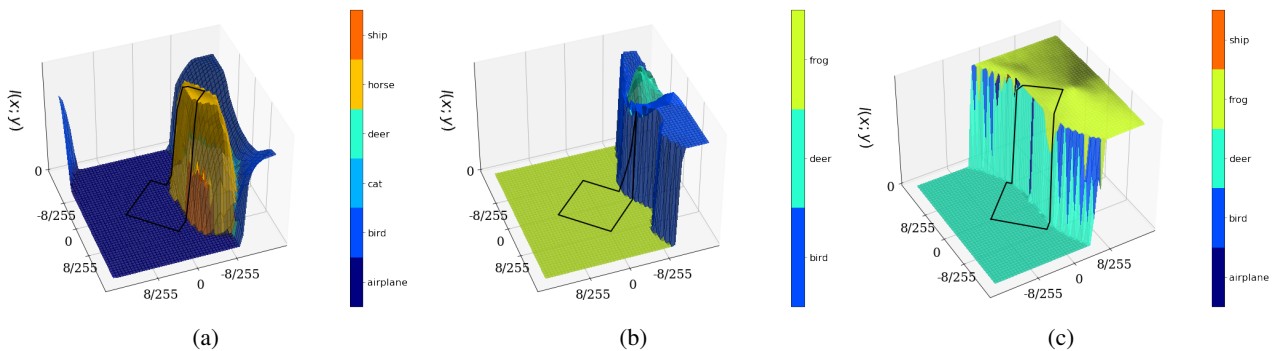

(a)  (b)  (c)

*Figure 10.* Cross-entropy loss surface produced by the highest-scoring CEB model (15.8% under AA+MT) for a test image of **(a)** an airplane, **(b)** a frog, **(c)** a deer. The diamond-shape represents the projected $l_\infty$ ball of size $\epsilon = 8/255$ around the original image. The surface is generated by varying the input to the model, starting from the original input image toward either the worst attack found using PGD or the one found using a random direction.