# OpenReview forum: "A Closer Look at the Adversarial Robustness of Information Bottleneck Models"
_ICML.cc/2021/Workshop/AML — ICML 2021 Workshop AML Poster_

### Official Review · Reviewer_rg4g · 2021-06-20
**The authors provide a possible explanation of the robustness of IB models through experiment results.**

**Rating:** Accept
**Confidence:** 4

**Review:**

- The authors study the reason why information bottleneck (IB) models tend to exhibit stronger robustness under adversarial attack. Through conducting adversarial experiments on CIFAR-10 and MINIST with VIB and CEB models under FGSM and PGD attack, the authors attribute the robustness of IB models to gradient obfuscation.
- However, the explanation seems to be a little arbitrary. For example, in MNIST experiment, the effectiveness of PGD than FGSM doesn't necessarily results from gradient obfuscation. Also, it would be more persuasive if theoretical analysis is provided besides empirical results. But generally speaking, this paper is a good attempt for explaining the robustness of IB models.

Also there is a little typo on the left side of line 36:

`also know as -> also known as`

---

### Decision · Program_Chairs · 2021-06-21

**Decision:**

Accept (Poster)

**Comment:**

This paper studied the robustness of information bottleneck models. The reviewer raised concerns about the explanation. The authors are encouraged to further address the concerns.